# EGFR R521K Polymorphism Is Not a Major Determinant of Clinical Cetuximab Resistance in Head and Neck Cancer

**DOI:** 10.3390/cancers14102407

**Published:** 2022-05-13

**Authors:** Mihály Cserepes, Györgyi A. Nelhűbel, Mónika Meilinger-Dobra, Adrienn Herczeg, Dóra Türk, Zita Hegedűs, Laura Svajda, Erzsébet Rásó, Andrea Ladányi, Kristóf György Csikó, István Kenessey, Árpád Szöőr, György Vereb, Éva Remenár, József Tóvári

**Affiliations:** 1Department of Experimental Pharmacology, National Institute of Oncology, Ráth György utca 7-9, H-1122 Budapest, Hungary; cserepestm@oncol.hu (M.C.); nelhubel.gyorgyi@med.semmelweis-univ.hu (G.A.N.); turkdora79@gmail.com (D.T.); hegeduszita@yahoo.com (Z.H.); svajda.laura@ext.oncol.hu (L.S.); 2National Tumor Biology Laboratory, National Institute of Oncology, H-1122 Budapest, Hungary; ladanyi@oncol.hu (A.L.); kenessey.istvan@oncol.hu (I.K.); 3The Multidisciplinary Head and Neck Cancer Center, National Institute of Oncology, H-1122 Budapest, Hungary; moni.dobra@gmail.com (M.M.-D.); herczeg.adrienn@oncol.hu (A.H.); remenar.eva@irgalmas.hu (É.R.); 4Department of Pathology, Forensic and Insurance Medicine, Semmelweis University, H-1091 Budapest, Hungary; raso.erzsebet@med.semmelweis-univ.hu; 5Department of Surgical and Molecular Pathology, National Institute of Oncology, H-1122 Budapest, Hungary; 6Department of Chest and Abdominal Tumors and Clinical Pharmacology, National Institute of Oncology, H-1122 Budapest, Hungary; csiko.kristof.gyorgy@oncol.hu; 7Hungarian Cancer Registry, National Institute of Oncology, H-1122 Budapest, Hungary; 8Department of Biophysics and Cell Biology, Faculty of Medicine, University of Debrecen, H-4032 Debrecen, Hungary; akuka@med.unideb.hu (Á.S.); gvereb2020@gmail.com (G.V.)

**Keywords:** EGFR, head and neck cancer, cetuximab, drug resistance, R521K, antibody-dependent cellular cytotoxicity

## Abstract

**Simple Summary:**

Malignant Head and neck squamous cell carcinomas occur frequently, and several treatment regimens are used to fight disease progression. While anti-Epidermal growth factor receptor (EGFR) antibody cetuximab is applied successfully in many cases, therapy resistance occurs after a short period in numerous patients. We checked the hypothesis whether EGFRvIII or EGFR R521K variants can be responsible for antibody efficacy or therapy resistance. EGFRvIII, unlike stated before, was found extremely rarely (<1%), while EGFR R521K was present in over 40% of the patients and suggested to be important in the preclinical models, but not in the clinical cohort. Conclusively, our results suggest that neither EGFRvIII nor EGFR R521K variants are directly resulting cetuximab resistance in Head and neck squamous cell carcinoma patients.

**Abstract:**

Background: Head and neck squamous cell carcinomas (HNSCCs) are among the most abundant malignancies worldwide. Patients with recurrent/metastatic disease undergo combination chemotherapy containing cetuximab, the monoclonal antibody used against the epidermal growth factor receptor (EGFR). Cetuximab augments the effect of chemotherapy; however, a significant number of patients show therapy resistance. The mechanism of resistance is yet to be unveiled, although extracellular alterations of the receptor have been reported, and their role in cetuximab failure has been proposed. Aims: Here, we investigate possible effects of the multi-exon deletion variant (EGFRvIII), and the single nucleotide polymorphism EGFR R521K on cetuximab efficacy. Results: Our results show that in HNSCC patients, the EGFRvIII allele frequency is under 1%; therefore, it cannot lead to common resistance. EGFR R521K, present in 42% of the patients, is investigated in vitro in four HNSCC cell lines (two wild-type and two heterozygous for EGFR R521K). While no direct effect is found to be related to the EGFR status, cells harboring R521K show a reduced sensitivity in ADCC experiments and in vivo xenograft experiments. However, this preclinical difference is not reflected in the progression-free or overall survival of HNSCC patients. Furthermore, NK cell and macrophage presence in tumors is not related to EGFR R521K. Discussion: Our results suggest that EGFR R521K, unlike reported previously, is unable to cause cetuximab resistance in HNSCC patients; therefore, its screening before therapy selection is not justifiable.

## 1. Introduction

Malignancies of the head and neck region annually affect more than 880,000 people worldwide, causing over 450,000 deaths every year [1]. The majority of the cases are represented by squamous cell carcinomas of the oral cavity, oropharynx, hypopharynx, and larynx (head and neck squamous cell carcinomas, HNSCCs). For surgically irresectable tumors, various (chemo)radiotherapy (CRT or RT) and chemotherapy regimens have been established. Traditionally, cisplatin was the cornerstone of the systemic treatment of HNSCC. However, different combination therapies including cisplatin have proved to act more efficiently than monotherapies. These approaches often involve cisplatin/carboplatin, 5-FU, paclitaxel, docetaxel as cytostatic/cytotoxic modalities, and cetuximab, the monoclonal antibody used against the epidermal growth factor receptor (EGFR) [2]. Additionally, recent advances have shown promising results, with novel checkpoint inhibitor therapeutics such as pembrolizumab [3] or nivolumab [4]; however, these immunotherapies are mostly available to patients with PD-L1-positive HNSCC; therefore, they do not providing a suitable alternative for many patients [5,6].

Combination chemotherapy, even if its superior effect over cisplatin monotherapy is debated [7], can be applied as selective first-line therapy combined with radiotherapy. Furthermore, combination therapy in the palliative treatment of recurrent/metastatic HNSCCs is widespread, involving cisplatin and 5-FU in combination with cetuximab for six weeks, followed by cetuximab monotherapy until disease progression (EXTREME protocol, EPF therapy) [8].

Despite the routine use of cetuximab, its beneficial effect seems to vary among patients. Up to now, no precise mechanism of cetuximab hypersensitivity or resistance was confirmed, although many theories, such as tumor hypoxia, tumor immunoevasion, tumor microbiome, and genetic alterations, have been broadly investigated [9].

Cetuximab, first described inhibiting EGFR in 1988 [10], was accepted for clinical use in the KRAS wild-type colorectal cancer (CRC) treatment, leading to the first routine genotyping of CRC patients, since somatic mutations of KRAS were found to confer cetuximab resistance [11]. Similar testing is necessary before the second-line cetuximab treatment of NSCLC patients too [12]. However, in HNSCC, RAS mutations occur rarely, indicating the role of different resistance mechanisms against anti-EGFR therapies. It was proposed earlier that hypoxia-induced epithelial–mesenchymal transition (EMT) might be responsible for cetuximab resistance [13], while others reported that the EMT process is initiated by exosomes, and cetuximab therapy might effectively suppress it [14]. The PI3K-AKT-mTOR molecular signaling axis is also suspected of playing a critical role in the cetuximab therapy response [15,16,17]. The critical role of immune suppression affecting cytotoxic T lymphocytes and NK cell activity was also proposed [18]. However, despite the multiple aspects, no conclusive results have yet been presented.

The structure of the target receptor EGFR and its alterations can also dramatically affect antibody binding and receptor inhibition. In colorectal cancer, EGFR S468R substitution induces cetuximab resistance; however, the newer-generation anti-EGFR antibody necitumumab can overcome this resistance and inhibit altered EGFR [19]. In HNSCC, the alterations of the extracellular domain were also investigated. A multi-exon deletion (EGFRvIII) was reported to be abundant in HNSCC [20], as well as single nucleotide polymorphism R521K [21]. A recent study claimed that polymorphism R521K was a negative predictive factor and the actual mediator of cetuximab resistance [22], with another paper even naming it a prognostic factor [23], while a third similar study found only a statistically non-significant trend in progression-free survival (PFS), and no difference in overall survival (OS) [24].

Our aim in the present study is to further delineate the significance of the presence of different EGFR alterations and immunophenotypes on cetuximab resistance in vitro, in vivo, and in the clinical follow-up of a retrospective cohort of recurrent/metastatic HNSCC patients, following the first-line palliative systemic treatment according to the EXTREME protocol, in order to clarify the importance of EGFR genotyping in the diagnostic process.

## 2. Materials and Methods

### 2.1. Chemicals

Cetuximab (Erbitux) was purchased from Merck KGaA (Darmstadt, Germany). All other necessary general chemicals were purchased from Sigma-Aldrich (St. Louis, MO, USA).

### 2.2. Cell Cultures

PE/CA-PJ15 (hereafter PJ15) and PE/CA-PJ41 (hereafter PJ41) human HNSCC cells (Sigma-Aldrich, St. Louis, MO, USA) were cultured in Iscove’s modified Dulbecco’s medium (Lonza, Basel, Switzerland), while Cal-27 and FaDu HNSCC cells (ATCC) were cultured in Dulbecco’s Modified Eagle’s Medium (Lonza), both supplemented with 10% fetal bovine serum (FBS) (BioSera, Boussens, France) and 1% penicillin/streptomycin (P/S) (Lonza). CD16.176V.NK-92 cells were cultured as previously described [25] (see Appendix A). All cells were controlled to be free from mycoplasma infection, and were cultured for no more than 25 passages/60 days after thawing.

### 2.3. Genetic Characterization of HNSCC Cell Lines and Patient Samples

We isolated total RNA from cell lines using the Direct-Zol RNA miniprep kit (Zymo Research) and genomic DNA using the NucleoSpin Tissue mini kit for DNA from cells and tissue kit (Macherey-Nagel). Formalin-fixed paraffin-embedded (FFPE) samples from HNSCC patients were also analyzed, total RNA and DNA content was obtained using High Pure FFPET RNA/DNA isolation kits (Roche, Basel, Switzerland). All procedures were performed strictly following the protocols from the manufacturers (for primers and details, see Appendix A).

All sanger sequencing data were analyzed, and samples where R521K mutant allele had a frequency of 50% or more (on the sequenograms of Sanger-sequencing) were considered as mutant cell lines or mutant clinical tumors. For EGFRvIII, the wild-type and mutation-specific PCR products were evaluated following gel electrophoresis. If a suspected EGFRvIII band was spotted, the DNA was purified and sent for sequencing to confirm genotype.

### 2.4. Cell Proliferation Assay

Cells were trypsinized and counted; then, 5 × 10^3^ cells were plated in each well of a 96-well plate. After cell attachment, cells were treated by addition of serially diluted stocks of cetuximab or cisplatin. Cell viability was measured using the MTT assay, as previously described [26].

### 2.5. Flow Cytometry

EGFR expression was measured with immunolabeling using mouse monoclonal antibody mAb 528 (isolated from the supernatant of the hybridoma ATCC HB-8509), with a slightly different binding site [27] and cetuximab (Merck). Alexa Fluor 647-conjugated secondary antibodies (goat-anti-mouse (GAMIG) and goat-anti-human (GAHIG), Invitrogen, Carlsbad, CA, USA) were used. The 10^6^ cells were labeled with 10 µg/mL of antibodies for 10 min on ice in PBS supplemented with 5 mM glucose. After each incubation, cells were washed twice by centrifugation at 400× *g*. At least 10,000 cells per sample were analyzed with an Attune NxT Acoustic Focusing Cytometer (Applied Biosystems, Life Technologies, Carlsbad, CA, USA).

### 2.6. EGFR Phosphorylation Assay

Cell cultures (10^6^ cells/flask) underwent cetuximab treatment (100 μM, 24 h). Cellular activation was induced using 5 nM EGF treatment on each sample for 30 min before harvesting. For sample processing and the detection of the EGFR phosphorylation status, we used the EGFR Phosphorylation Array kit (Raybiotech Inc., Peachtree Corners, GA, USA), according to the manufacturer’s protocol. The luminescent signals of the membranes were recorded using a digital imaging system (UVITEC, Cambridge, UK).

### 2.7. Antibody-Dependent Cellular Cytotoxicity Assay

Electric Cell-substrate Impedance Sensing (ECIS^®^; Applied BioPhysics, Inc., New York, NY, USA) was used to perform the kinetic analysis of in vitro killing mediated by CD16.176V.NK-92 cells (ADCC) as described [28,29]. Effector/target ratio was 1:1, and cetuximab was added at 1 µg/mL. Treatment started 25 h after seeding. Control, effector cell only, cetuximab only, and cetuximab with effector cell groups were compared co-temporally, in technical replicates, and repeated twice. Impedance was monitored for 24 h. Averaged traces were normalized to impedance measured at the start of treatment; then, impedances at the end of the 48 h time course were normalized to the corresponding value of the untreated co-culture control.

### 2.8. Animal Experiments

We inoculated 10^6^ tumor cells subcutaneously in 12-week-old female SCID (CB17/Icr-*Prkdc^scid^*/IcrIcoCrl) mice, on the right dorsal side. Tumor cell inoculation was performed on animals anesthetized with an intraperitoneal injection of a mixture of zolazepam (20 mg/kg), xylazine (12.5 mg/kg), butorphanol (3 mg/kg), and tiletamine (20 mg/kg). After the tumors reached 50 mm^3^ volume, the mice were treated with cetuximab according to the clinical regimen: 400 mg/m^2^ induction dose twice on the first week, followed by 250 mg/m^2^ dose intraperitoneally twice a week. Tumor growth was measured using a caliper, and tumor volumes were calculated as follows: length × width^2^/π. Animals with tumors reaching 2000 mm^3^, or after three weeks, were euthanized.

### 2.9. Clinical Data Analysis

We followed up with HNSCC patients treated at the National Institute of Oncology, Hungary, between 2011 and 2019, with EPF combination therapy (cetuximab + cisplatin/carboplatin + 5-FU) following the EXTREME protocol of palliative care of recurrent/metastatic disease. Of 117 HNSCC patients with known EGFR genotype and full medical records provided to us anonymously, data from 103 HPV-negative patients were included in the analyses. Objective response rates (complete or partial remission), disease control rates (complete or partial remission and stable disease), and progression-free and overall survival were the primary outcomes. We examined clinical outcome comparing EGFR wild-type and EGFR R521K patient groups.

### 2.10. Immunohistochemistry

Paraffin-embedded samples were sectioned and labeled for CD68, NKp46, and CD16, as described previously [30]. All samples were analyzed by two independent experts, scoring labeled cell density semiquantitatively between 0 and 3, and the mean value of their separate scores was used in the analysis.

### 2.11. Statistical Analysis

Significance of difference among cell lines was analyzed using one-way ANOVA with post hoc Tukey’s method. For survival data, log-rank test was used for comparison of cumulative risk for progression or death. Non-parametric data of first response were correlated with EGFR status using Mann–Whitney U-test. The various potential factor correlations were measured using Chi-squared tests. Analyses were performed using Statistica 13 software (Tibco Software Inc., Palo Alto, CA, USA).

## 3. Results

### 3.1. EGFRvIII and R521K Genotypes

The extracellular region of the EGFR is responsible for cetuximab binding and receptor activation; therefore, its genetic analysis could be essential for understanding the differences in cetuximab therapy efficacy.

The EGFRvIII analysis showed that all four HNSCC cell lines were wild-type. Contrary to previous publications, in our cohort, only 1 patient (of 116 successful RNA isolations) was found positive for EGFRvIII extracellular deletion, excluding the possibility that this variation could be responsible for a heterogeneous therapy response in HNSCC patients.

As for the EGFR R521K polymorphism, we found cell lines PJ41 and Cal-27 expressing wild-type EGFR only, while PJ15 and FaDu were heterozygous for the R521K polymorphism. In clinical samples of 117 HNSCC patients, we found that 68 (58%) patients were harboring wild-type EGFR, 39 (33%) had the heterozygous genotype, while 10 (9%) were homozygous for EGFR R521K.

The cell line genotypes allowed us to perform in vitro and in vivo experiments in order to test whether cetuximab efficacy might be related to different EGFR R521K statuses. As human papillomavirus (HPV) is a prognostic factor in HNSCC, we included 103 HPV-negative patients (of total 117) in our clinical cohort.

### 3.2. EGFR Expression Variances among the Used HNSCC Cell Lines

We quantified EGFR expression of the four cell lines using fluorescent labeling by anti-EGFR antibody mAb 528 and cetuximab. The quantitative results showed no significant difference in EGFR expression among the four cell lines when labeled with monoclonal antibody mAb 528 (Figure 1A). However, repeated measurements with cetuximab labeling showed that FaDu cells could bind significantly less cetuximab, and also the other heterozygous cell line, PJ15 tended to have less binding capacity. These results suggested that the quantity of EGFR may be similar in wild-type and R521K polymorphic cells, but did not exclude the possibility that binding the avidity of cetuximab to polymorphic EGFR might be different (Figure 1B).

### 3.3. Cetuximab Sensitivity of HNSCC Cells in Vitro

Based on the genotypes, we expected differences in the cetuximab sensitivity of the cells. To clarify this possible effect, we performed proliferation assays to see the direct growth inhibition/toxicity of cetuximab on the cells. Up to a 100 μM concentration, none of the cell lines showed any dose-dependent sensitivity to the treatment (Figure 2A).

For the examination of the effects of cetuximab treatment on EGFR activity, we used a multi-target, protein-based array to follow EGFR, and nine different phospho-EGFR protein levels. The results showed that all four HNSCC cell lines showed a dramatic drop in EGF-induced EGFR phosphorylation when treated with cetuximab (phosphorylations on Tyr845, Tyr1173, and Ser1070 were the strongest without cetuximab treatment) (Figure 2B). This encouraged the hypothesis that direct effects on the tumor cells were unlikely to mediate differential tumor responses.

In vivo, the antibody cetuximab might lead to cell death not only by direct toxicity or signaling inhibition, but also by antibody-dependent cellular cytotoxicity (ADCC). We used the human NK-derived cell line for the ADCC experiments. Our results showed that, over the course of 24 h, EGFR wild-type cells (PJ41, Cal-27) were more sensitive to the cetuximab + NK cell treatment compared to the effect of NK cells only, while in R521K cell lines (PJ15, FaDu), only a minimal effect was seen, as the tumor cell viability (compared to co-culture without cetuximab) was 87% after 24 h, enabling the EGFR R521K polymorphism to potentially jeopardize cetuximab-mediated ADCC (Figure 3). Interestingly, the cetuximab-driven toxicity was more pronounced in PJ41 than in Cal-27, suggesting that other factors might play an important role in cetuximab efficacy.

### 3.4. Human HNSCC Cell Line-Derived Xenograft Growth

In order to investigate the possible effect of EGFR R521K on the cetuximab therapy success, we used subcutaneous murine xenograft models of the four different human HNSCC cell lines in SCID mice. All animals received cetuximab or physiological saline twice a week from the day the tumors reached 50 mm^3^. We followed up the tumor growth for three to five weeks, according to tumor growth. In EGFR wild-type PJ41 and Cal-27, the cetuximab treatment caused a nearly complete response, and the tumors entered into remission. EGFR R521K harboring FaDu xenografts did not shrink, but the tumor growth was efficiently inhibited. Interestingly, PJ15 tumor growth was barely affected by cetuximab (Figure 4). These results suggested that while the treatment seemed to be successful against tumors with wild-type EGFR, it was also effective in one model harboring EGFR R521K, suggesting that other factors might contribute to cetuximab therapy efficacy.

### 3.5. Immunophenotype in Wild-Type and EGFR R521 Tumor Samples and Its Correlation with Clinical Cetuximab Efficacy

To characterize the presence of possible effector cells mediating ADCC in HNSCC tumors, we examined the NK cells and macrophages in 78 samples of our clinical HNSCC cohort using immunohistochemistry. The staining score of cells expressing CD16 (FcγRIII), CD68 (macrophage marker), and NKp46 (NK cell marker) were used to quantify the presence of potential ADCC effector cells in the tumors (Figure 5). We examined whether EGFR R521K was possibly influencing the density of ADCC effector cells (thus, having therapy response potential) of the patients. No correlations were found between the EGFR status and pre-treatment immune status. This implied that no differences in the intratumoral CD16+ cell, NK cell, or macrophage status were present before the treatment; thus, the mutation did not lead to an immunocompromised environment. Additionally, the intratumoral density of the examined immune cells was not associated with the extent of progression-free or overall survival. Of note, the routine clinical sample collection was limited to samples taken before the treatment, as no samples were available following cetuximab therapy.

### 3.6. Clinical Impact of EGFR R521K Status on Cetuximab Therapy Outcome

We analyzed the clinicopathological factors in our cohort of 103 HNSCC patients (after excluding fourteen HPV-positive cases). All patients received EPF induction chemotherapy. We categorized the patients by the EGFR R521K status, considering R521K and all those which expressed at least 50% of the mutant allele according to the sequenograms. The analysis of correlations between clinical features of the patients and the EGFR R521K status (summarized in Table 1) revealed that there was a significantly higher ratio of women among EGFR R521K mutant cases than in wild-type cases. From another perspective, the occurrence of the mutation among women was 64% (14 of 22 cases), while in men, only 33% (27 of 81 cases). However, other investigated clinicopathological features did not show any correlation with the EGFR status of the patients.

We compared the progression-free and overall survival of wild-type and R521K mutant patients, concluding that the EGFR R521K status was neither predictive nor prognostic in our HNSCC cohort (Figure 6). The stratification of the survival data by sex did not change the results, alleviating the concerns whether an uneven sexual distribution could bias the results.

## 4. Discussion

Anti-EGFR antibody cetuximab therapy has successfully become part of the standard systemic therapy regimen of recurrent/metastatic HNSCC [8]. Due to the occurrence of unsatisfactory therapy responses, the identification of cetuximab-resistant patients is of high importance. The mechanism(s) of resistance are still unclear; however, alterations of EGFR might be involved. The abundance of the extracellular alterations of EGFR in HNSCC was debated. In our study, we clarified EGFRvIII and EGFR R521K statuses in four HNSCC cell lines and in 117 HNSCC patients undergoing cetuximab-containing chemotherapy. Our results showed that EGFRvIII was barely present in our samples (one patient only), while EGFR R521K was found to be present with at least one allele in 42% of the patients. Importantly, former reports varied upon the EGFRvIII rate in HNSCC, one study finding the allele frequency to be as common as 42% [20]. Along with other studies [31,32], our data sharply contradicted this and called for the critical consideration of published data. These results serve as evidence that the EGFRvIII variant cannot be responsible for a high proportion of cetuximab-refractory tumors in HNSCC patients.

Our cellular assays of viability and EGFR phosphorylation inhibition showed no specificity of direct cellular effects of the anti-EGFR antibody, while cetuximab binding tests and the in vitro ADCC experiments were in line with the theory that cetuximab might be more effective against EGFR 521 wild-type HNSCC cells. Our in vivo experiments also reflected a difference in cetuximab response among the xenograft models, with wild-type EGFR models (PJ41, Cal-27) showing a complete response, while the models carrying the EGFR R521K polymorphism showed a diverse response (but less than the wild-type models). Markedly, FaDu was quite sensitive to the cetuximab treatment, suggesting a multi-factor mechanism underlying the cetuximab response of the tumors. The ADCC and xenograft experiments both showed a differential efficacy, and highlighted the importance of NK cells in the antitumor efficacy of cetuximab. It is worth remarking that the immunodeficient SCID mice still possessed functional NK cells and macrophages, which could contribute to the response of the HNSCC xenografts. Despite the logical assumptions based on our preclinical models, clinical data of our HNSCC patient cohort did not show any patterns of immune status related to survival parameters. This could be due to a methodological issue, as the immune status of the patient samples was screened before the cetuximab treatment, and did not reflect any cetuximab-induced ADCC response differences. At present, there is no routine sampling of HNSCC tumors after cetuximab treatment, which prohibits the analysis of such data. To address this issue, an analysis of tumor samples taken following the treatment would be favorable. Our cohort of 103 HPV-negative HNSCC patients with recurrent/metastatic disease, compared to the original EXTREME report, had longer progression-free and overall survival (for PFS: 29 vs. 24 weeks; for OS: 50 vs. 43 weeks). Similarly, the overall response rate (57% vs. 40%) and disease control rate (86% vs. 81%) were higher in our cohort [8]. Despite the longer timespan, however, we did not observe any significant predictive or prognostic effect of EGFR R521K polymorphism. Studies which found a slight significance in patient response [22] or prognosis [23], or like our cohort did not show any significance of the EGFR status on clinical outcome [24], along with our results, might be subject for a meta-analysis; however, it is unlikely that the large variation in patient response and patient survival would rely on the R521K polymorphism as a single determining factor.

Of note, preclinical models of cetuximab therapy resistance are often difficult to translate to patient care, as cetuximab is only one of the combined compounds used in the treatment of recurrent/metastatic HNSCC. Moreover, cell line models can harbor more differences not investigated, which might enhance effects in preclinical models. The higher number of cell lines used in our study and by others [22] partially evades this problem, but cell-line based results are always better to correlate with clinical data. The clinical setup enables a much longer timespan to examine differences, and, importantly, reflects the particular patient cohort status, either their general health, or special genetic patterns they share. Therefore, we suggest that reports from different populations are beneficial to be reported publicly, making a comparison possible, which could lead to potential new predictive pattern recognition.

We propose that the focused investigation of tumor cell traits along with the immune features of the tumor is unavoidable in order to understand the drivers of cetuximab resistance, and to choose patients most eligible for the EGFR-targeting therapy regimen. On the other hand, research aiming to improve antibody treatment efficacy has already led to promising combination regimens [33] and further generations of anti-EGFR antibody therapeutic possibilities, which might overcome the therapy resistance of HNSCC tumors [19,22]. In the case of the clinically used cetuximab, however, our data did not support the use of EGFR extracellular domain alterations as a predictor of clinical success of therapy.

## 5. Conclusions

The background of clinically observed resistance to anti-EGFR antibody cetuximab in HNSCC patients is unknown. Extracellular modifications of the receptor might alter antibody affinity and, consequently, therapy efficacy. Immune cell presence in the tumor is also a determinant of a possible antitumor immune response.

In our study, we examined these factors in vitro, in vivo, and on clinical samples. While in vitro and in vivo data from cell lines suggested that single nucleotide polymorphism R521K might reduce cetuximab binding, the antibody-dependent cytotoxicity, and in vivo antitumor effect of cetuximab, the clinical data of 103 patients showed no difference in therapy response and progression-free or overall survival. Clinical outcome was also independent of pre-treatment NK cell and macrophage presence in the HNSCC tumor tissue.

Our data help to elucidate the role of receptor alterations in clinical cetuximab resistance.

## Figures and Tables

**Figure 1 cancers-14-02407-f001:**
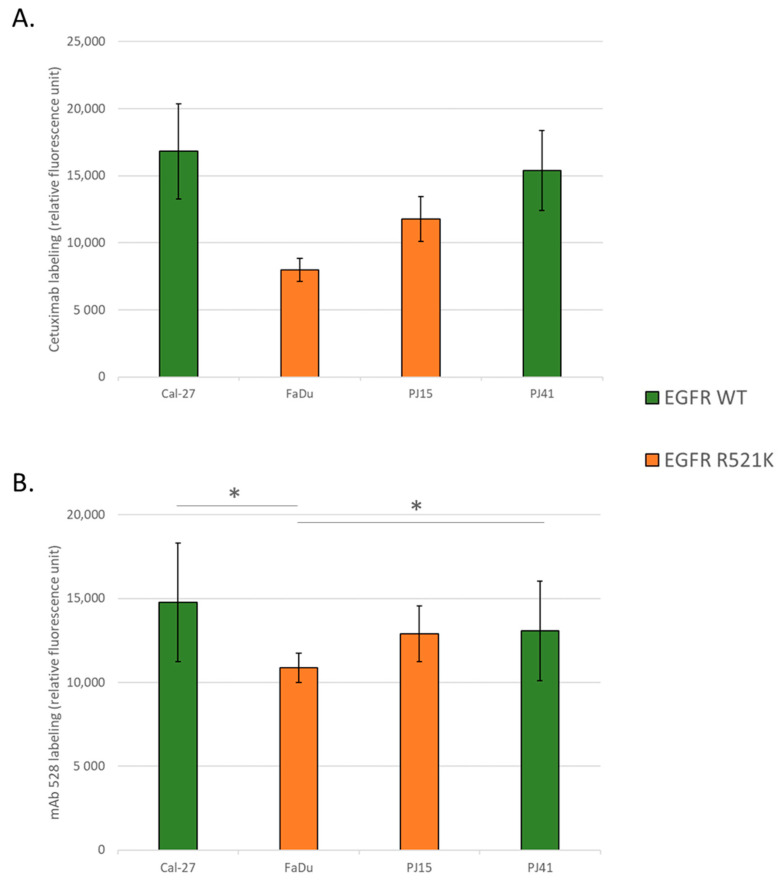
EGFR expression level in HNSCC cells with different EGFR R521K statuses. EGF receptor expression was established using labeling with mouse monoclonal antibody mAb 528 (**A**). Cetuximab-binding capacity of the cells was measured after labeling with cetuximab (**B**). Expression values are shown in relative fluorescence units (RFUs) obtained in flow cytometry measurements. All bars represent mean ± SD values of three parallel measurements. Statistical significance of one-way ANOVA with Tukey’s post hoc test is represented. *: *p* < 0.05.

**Figure 2 cancers-14-02407-f002:**
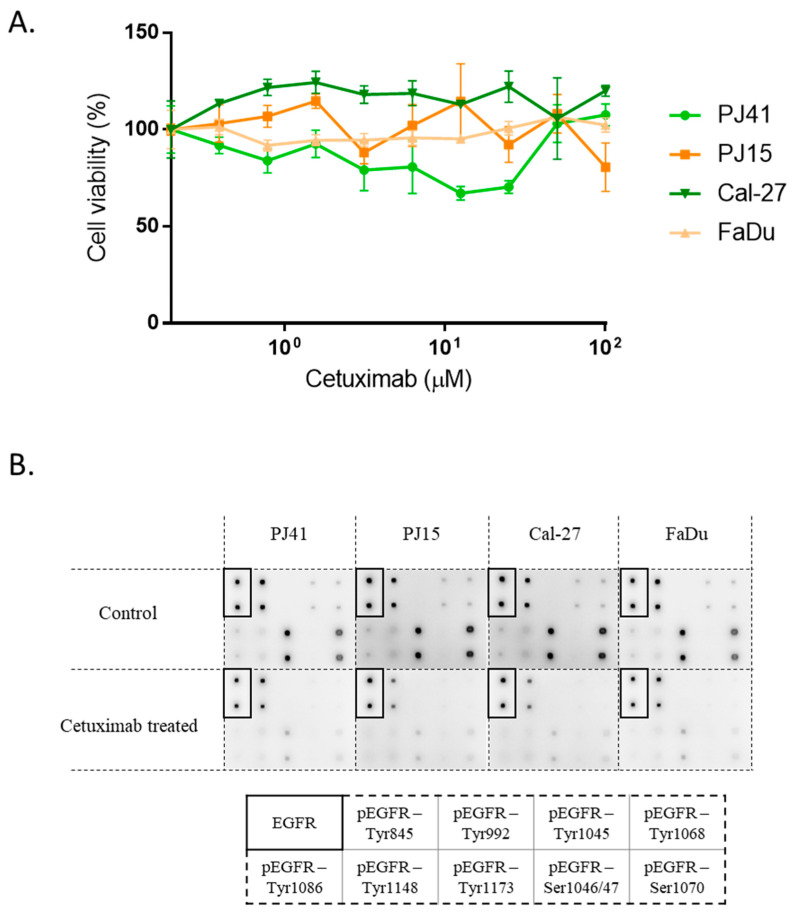
Cetuximab was not selective on HNSCC cells in vitro. (**A**) Cetuximab treatment was not toxic on HNSCC cell lines in vitro. Cell viability was quantified using MTT colorimetric assay (mean ± SD, *n* = 3). (**B**) Cetuximab treatment caused general decrease in EGFR activation at nine phosphorylation sites regardless of EGFR R521K status.

**Figure 3 cancers-14-02407-f003:**
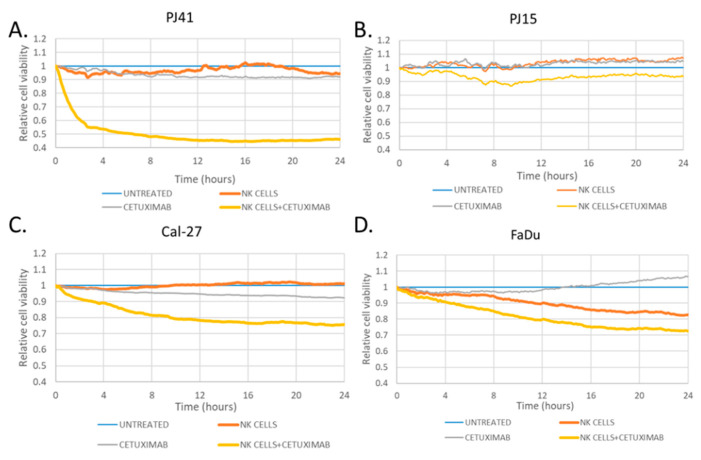
Reduced antibody-dependent cellular cytotoxicity on EGFR R521K cells. All cells were followed up for 24 h in the ECIS system to record cell death in untreated or cetuximab-treated tumor cells in the presence or absence of human NK cells. PJ41 (**A**) and Cal-27 (**C**) showed high or moderate cetuximab sensitivity, compared to NK cell co-culture without treatment (yellow vs. orange curves, 49% and 74% of the cells were alive, respectively). PJ15 (**B**) and FaDu (**D**) showed lower sensitivity (87% of the cells were alive in both cell lines after 24 h of cetuximab treatment).

**Figure 4 cancers-14-02407-f004:**
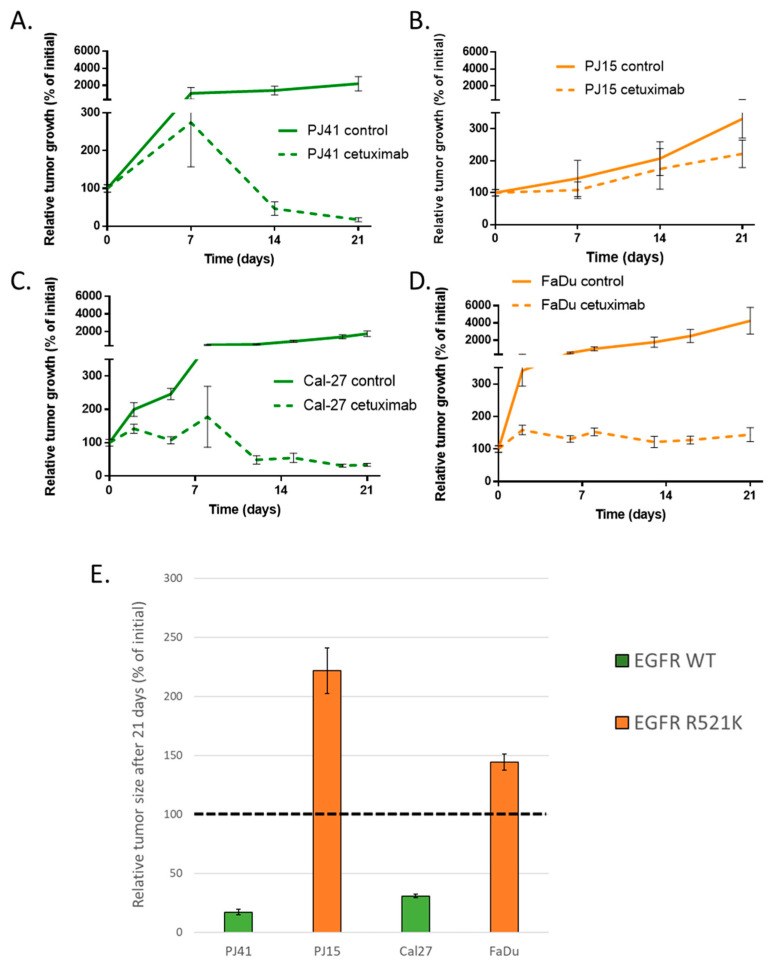
The in vivo antitumor effect of cetuximab was different among the cell models. Subcutaneously growing tumor xenografts were treated with cetuximab. In EGFR wild-type PJ41 (**A**) and Cal-27 (**C**) models, the treatment caused an approximately complete response, while in R521K mutant model PJ15 (**B**), the cetuximab-treated tumors grew. Interestingly, in FaDu tumors with EGFR R521K (**D**), cetuximab effectively inhibited tumor growth, but did not lead to tumor size reduction. (**E**) Relative tumor sizes in cetuximab-treated mice after 21-day course of treatment. Dashed line represents the tumor size (100%) at the first treatment. Control animals were treated with physiological saline i.p. All data represent mean ± SD. PJ41 and PJ15: *n* = 5. Cal-27 and FaDu: *n* = 7.

**Figure 5 cancers-14-02407-f005:**
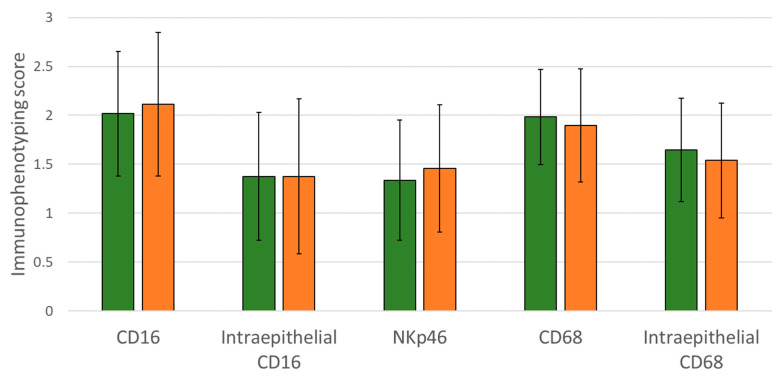
Immune status was independent of EGFR R521K status in HNSCC patients. Semiquantitative scoring of immunohistochemical labeling of CD16, NKp46, and CD68. In case of CD16 and CD68, intraepithelial staining was also evaluated. Bars represent mean ± SD of two independent counts.

**Figure 6 cancers-14-02407-f006:**
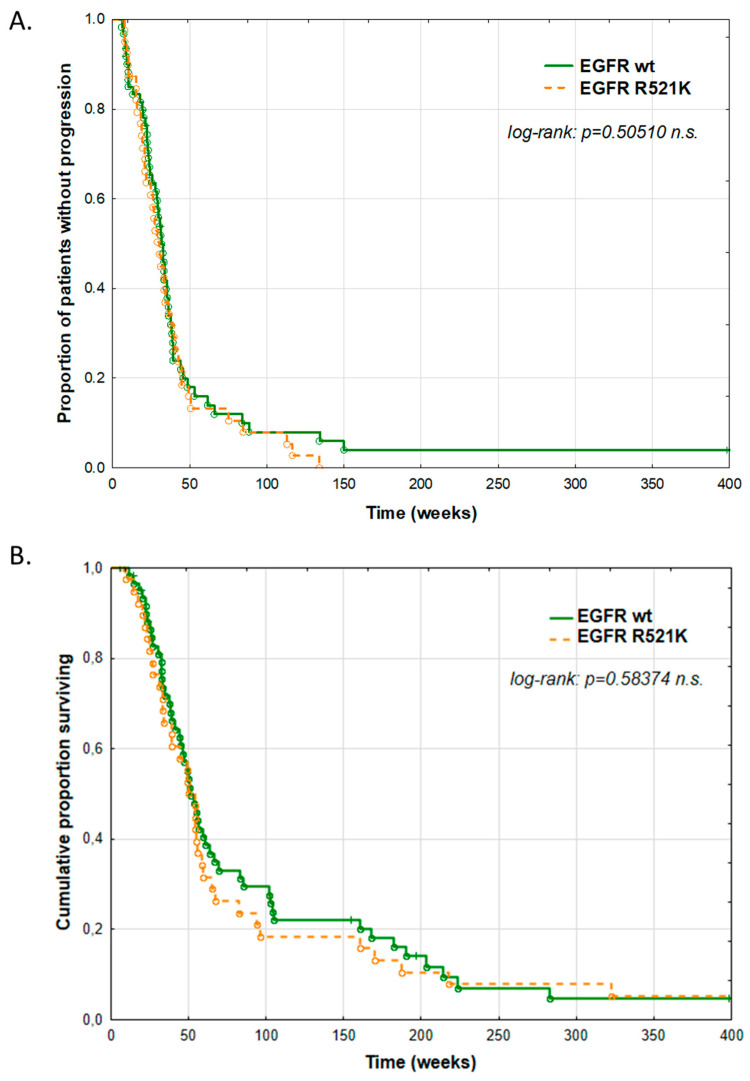
EGFR R521K status is not predictive nor prognostic in HNSCC patients. Clinicopathological data was collected for 103 HPV-negative HNSCC patients with known EGFR R521K status. Progression-free survival (**A**) and overall survival (**B**) of the patients with wild type or R521K (mutant allele frequency ≥ 50%) EGFR were analyzed using log-rank method.

**Table 1 cancers-14-02407-t001:** Clinicopathological features of the clinical HNSCC patient cohort.

Patients	Total 103	EGFR wt 62 (60%)	EGFR R521K 41 (40%)	*p*
Men (*n*, %) Women (*n*, %)	81 (79%) 22 (21%)	54 (87%) 8 (13%)	27 (66%) 14 (34%)	*p* < 0.05
Age (median, years)	59.5	60	58	No correlation
Overall response rate (*n*, %)	59 (57%)	39 (63%)	20 (49%)	No correlation
Disease control rate (*n*, %)	89 (86%)	52 (84%)	37 (90%)	No correlation
Progression-free survival (median, weeks)	29	29	28	No correlation
Overall survival (median, weeks)	50	48	49	No correlation

## Data Availability

Data are contained within the article or Appendix A. For patient data, no personal identification data are available.

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
