# Peer review of "EGFR R521K Polymorphism Is Not a Major Determinant of Clinical Cetuximab Resistance in Head and Neck Cancer"

_cancers, 2022, doi:10.3390/cancers14102407_

Round 1
Reviewer 1 Report
The manuscript is improved. Several English language errors remain, including the final sentence of the manuscript, which is nonsensical. Perhaps the authors meant elucidate rather than alleviate?
Author Response
We are thankful to the reviewer’s work to improve the quality of the manuscript. A thorough grammar and English language check was applied with native speaker, and several errors were corrected (all marked in the revised manuscript).
In the last sentence, indeed, our intention was to write that our data helps to clarify the role of these alterations in resistance. (Namely that the two particular alterations do not have severe clinical consequences). We revised and changed the wording „alleviate à elucidate”.
Thank you again to consider our paper for publication in Cancers.
Reviewer 2 Report
Authors have tried to present in the present manuscript that EGFR R521K polymorphism is not a major detriminant of clinical resistance to cetuximab in HNSCC. The authors have done quite a substantial work , but i have following comments ,
- In cell survival assay in figure 2, authors are showing total % of cells more than 100 % which is not correct .
- In addition to it none of the cell lines were characterized properly and supplementary results only show methodology for identification R1521K positive cells, but results are not shown . The karyotyping of cell lines are also not done .
- The authors are mentioning that they have followed the data of 103 patients. They have not shared any clearance form human ethical committee and animal ethical committee. There is no mention of how EGFR allelic mutation was identified in patients.
- The authors need to clarify , discuss and suggest that why the in vitro and invivo and patient results regarding R521K mediated cetuximab resistance is not same.
Author Response
We are thankful to the reviewer for the in-depth discussion of our current manuscript in order to improve its quality. Below, we address the four questions in particular, please see the attached file. We also modified the manuscript document accordingly, marking every changes.

This manuscript is a resubmission of an earlier submission. The following is a list of the peer review reports and author responses from that submission.
Round 1
Reviewer 1 Report
In this article, EGFR R521K, present in 54% of the patients, was investigated in vitro in four HNSCC cell lines. The authors showed that cetuximab caused complete response in EDFR wild type PJ41 and Cal-27 in vivo and cetuximab was not effectively inhibiting tumor growth in PJ15 and FaDu. Moreover, the authors showed that the preclinical difference was not reflected in progression-free, or overall survival of HNSCC patients.
minor comments
- I think that this study needs to be approved by the ethics committee because this study included clinical data analysis.
- Please change “indergo” to “undergo” in the abstract section.
- Please change “tumor sie reduction” to “tumor size reduction” in figure 4 legend.
Reviewer 2 Report
In this manuscript, the authors seek to determine if EGFR alterations are responsible for cetuximab resistance. They show that in their clinical cohort, the multiexon variant (EGFR vIII) is under 1%, suggesting that this variant does not lead to resistance. They do identify EGFR R521K mutations in HNSCC patients and characterize the properties of cell lines either WT or heterozygous for the allele. They show that cell lines heterozygous for R521K show reduced antibody independent cellular cytotoxity. In vivo xenograft experiments showed that tumor growth was not affected by cetuximab for one heterozygous cell line. However, patients with the R521K mutation did not differ significantly in progression free or overall survival from the WT patients. The results are interesting and provide helpful information to understanding cetuximab resistance, but the results appear preliminary. Major concerns are noted below:
- The clinical cohort should be better described in the manuscript and appears relatively small for the conclusions drawn in the manuscript. For example, the response rate and disease control rate in this cohort is higher than in other published studies, suggesting that cetuximab resistance was not common in this group, yet the authors use the low frequency of EGFR vIII in this cohort to exclude any contribution of this allele.
- 25 of the cohort patients were heterozygous for R521K and 9 were homozygous for EGFR R521K. The authors do not include a cell line homozygous for R521K nor do they separate patients according to heterozygous or homozygous status. Would this contribute to the effects?
- How many times were the experiments in Figure 1 performed? How was statistical significance determined for the results?
- It is striking that Cal-27 (homozygous) and FaDu (heterozygous) behave almost identically in these assays while PJ41 (homozygous) and PJ15 (heterozygous) respond very differently. The authors state that this suggests the effects are multifactorial, but do not develop any potential hypothesis on why the cell lines respond so differently. The authors do not prove that R521K responsible for the reduced sensitivity in the ADCC or xenograft experiments.
Minor concerns:
- The figure presentation should be improved. Axis and figure labeling are difficult to read.
- The presentation of data in the table is difficult to interpret with the authors’ points. The high rate of the 521K mutation in women cannot be ascertained from the total percentages.
- The manuscript contains many grammatical errors and should be extensively proofread